# C3T: Cross-modal Transfer Through Time for Human Action Recognition

## Abstract

In order to unlock the potential of diverse sensors, we investigate a method to transfer knowledge between modalities using the structure of a unified multimodal representation space for Human Action Recognition (HAR). We formalize and explore an understudied cross-modal transfer setting we term Unsupervised Modality Adaptation (UMA), where the modality used in testing is not used in supervised training, i.e. zero labeled instances of the test modality are available during training. We develop three methods to perform UMA: Student-Teacher (ST), Contrastive Alignment (CA), and Cross-modal Transfer Through Time (C3T). Our extensive experiments on various camera+IMU datasets compare these methods to each other in the UMA setting, and to their empirical upper bound in the supervised setting. The results indicate C3T is the most robust and highest performing by at least a margin of 8%, and nears the supervised setting performance even in the presence of temporal noise. This method introduces a novel mechanism for aligning signals across time-varying latent vectors, extracted from the receptive field of temporal convolutions. Our findings suggest that C3T has significant potential for developing generalizable models for time-series sensor data, opening new avenues for multi-modal learning in various applications.

## 1 Introduction

Humans can naturally actuate a motion they have only seen before; however, transferring motion knowledge across sensors for machine learning models is nontrivial. Our interaction with computing has historically been centered around visual and textual modalities, which has provided these models an abundance of data. Thus, deep learning based human action recognition (HAR) systems often collapse 3D motion into related but imprecise modalities such as visual data (Ji et al., 2012; Simonyan & Zisserman, 2014; Lin et al., 2022; Wang et al., 2023) or language models (Radford et al., 2021; Wang et al., 2022a; Tong et al., 2022; Piergiovanni et al., 2023; Feng et al., 2023).

Inertial Measurement Units (IMUs), which typically provide 3-axis acceleration and 3-axis gyroscopic information on a wearable device, emerge as strong candidates for understanding human motion in a nonintrusive fashion. Smartwatches, smartphones, earbuds and other wearables have enabled the seamless integration of IMUs into daily life (Mollyn et al., 2023). Unfortunately, IMUs remain underutilized in current machine-learning approaches due to several challenges including (1) the lack of abundant labeled data (2) the difficulty in interpreting the raw sensor readings, and (3) a low-dimensional representation, which limits the amount of information that can be stored and learned from.

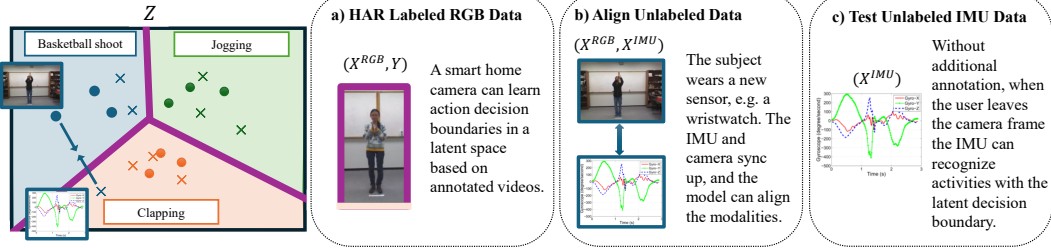

Figure 1: **Motivation for Unsupervised Modality Adaptation:** Depicts a scenario in which an AI system can add a new modality using no new labels, through latent space alignment.

Beyond IMUs, various sensing modalities are gaining popularity in wearables (e.g. surface electro-cardiogram, electromyography) and ambient monitoring systems (e.g. WiFi signals, Radar). While numerous AI methods have been developed for HAR using these modalities, they struggle to achieve the level of generalization and capability demonstrated by visual and textual modalities, which benefit from internet-scale data, and new modalities are continuously emerging. This raises the critical question of *how to integrate new sensors with existing ones in the absence of labeled data.* One promising solution is to leverage a well-documented modality to transfer knowledge to another modality, a process known as cross-modal transfer (Niu et al., 2020). Ideally, the model for this source modality would teach the model for the new target modality without additional human annotation effort. Existing cross-modal learning techniques assume a semi-supervised or fully supervised setup where there exists some labels for each modality during training (An et al., 2021; Niu et al., 2020). Cross-modal learning has not thoroughly been investigated in a setting where one modality is completely unlabeled during training, which we refer to as **Unsupervised Modality Adaptation** (UMA), demonstrated in Figure 1.

Multimodal learning and sensor fusion has long been studied for human action recognition, however, few methods study the UMA setting. In addition, existing works typically compress entire time-series sequences into single latent representations or segment data into fixed-size chunks for transformer inputs. However, this approach fails to account for the continuous nature of real-world data and the variable time spans over which actions occur. We propose 3 methods for the UMA setting. The first is called Student-Teacher (ST) akin to many existing knowledge distillation methods for other domain adaptation or semi-supervised settings. The second method, Contrastive Alignment (CA), aligns the latent representation of each data sample across modalities and uses a shared task head to perform transfer across modalities. The third method, entitled Cross-modal Transfer Through Time (C3T), extracts a time-varying latent dimension through the temporal receptive field of 3D convolution and uses a shared task self-attention head to perform transfer.

Our research evaluates these three UMA methods—ST, CA, and C3T—by focusing on the challenging task of transferring knowledge from RGB video data to IMU signals across four diverse human action recognition datasets. Cross-modal Transfer Through Time (C3T) surpasses the other methods on all datasets by at least an 8% margin on Top-1 Accuracy. Remarkably, C3T in unsupervised modality adaptation even outperforms the RGB unimodal supervised setting, underscoring its effectiveness in cross-modal learning.

Our qualitative analysis suggests that the success of CA and C3T lies in their ability to uncover hidden correlations between modalities within the latent space. This capability allows the model to leverage structural information from one modality to inform its understanding of another, even when labeled data is unavailable. To enhance performance over CA, C3T introduces a mechanism for aligning signals across time-varying latent vectors. This novel approach enables the model to capture temporal dynamics more effectively, resulting in enhanced robustness against time-related noise—a common challenge in real-world sensor data. We conduct additional experiments that demonstrate C3T's superior resilience to time-shift, misalignment, and time-dilation noise compared to other methods. The promising results of C3T open new avenues for developing highly generalizable models for time-series sensor data. This has far-reaching implications across various domains, including healthcare monitoring, smarthomes, industrial IoT, and human-computer interaction, where multi-modal learning from limited labeled data is crucial.

**Our contributions are as follows:**

- A novel motivation and setup for cross-modal transfer learning to an unlabeled modality, referred to as Unlabeled Modality Adaptation, UMA.
- Development and comparative analysis of three methods—Student-Teacher (ST), Contrastive Alignment (CA), and Cross-modal Transfer Through Time (C3T)—for performing UMA in human action recognition using RGB and IMU data (Sections 4 and 5)
- Insights into the mechanisms of latent space alignment that enable effective knowledge transfer between modalities, potentially applicable to other multi-modal learning tasks beyond human action recognition (Section 5).
- An in-depth analysis of how C3T's structure, in particular its use of temporal tokens, provides superior performance with minimal model size and enhanced robustness to temporal distortions, showcasing its potential for real-world applications with sensor data (Sections 5.1 and 5.2)

## 2 RELATED WORKS

We outline our motivation for the term UMA and then review the works related to each of our methods: Student-Teacher, Contrastive Alignment, and Cross-modal Transfer Through Time.

**Unsupervised Modality Adaptation** Cross-modal transfer learning is a powerful technique that enables the transfer of knowledge from a data-rich modality to one with limited data availability (Niu et al., 2020). This approach can be viewed as a specialized form of domain adaptation, where a model, trained in a source domain, is tasked with efficiently adapting to a related target domain for the same output task, despite having access to fewer labeled data points (Pan & Yang, 2009; Farahani et al., 2021). Given the focus on domains with scarce labels, adaptation is often achieved through unsupervised (Chang et al., 2020) or semi-supervised (An et al., 2021) methods. In the context of human activity recognition (HAR), domain adaptation may involve adapting between different sensors (Bhalla et al., 2021), adjusting to varying positions of wearables on the body (Wang et al., 2018; Chang et al., 2020; Prabono et al., 2021), accommodating different users (Hu et al., 2023; Fu et al., 2021), or adapting to different IMU device types (Khan et al., 2018; Zhou et al., 2020; Chakma et al., 2021). Our work focuses on domain adaptation where the target domain is a new completely unlabeled modality, hence we introduce the term Unsupervised Modality Adaptation (UMA). Previous research explored similar concepts under various terms like knowledge distillation, missing modality, robust sensor fusion, multi-modal alignment, and zero-shot cross-modal transfer. We propose UMA as an encompassing term to unify the discussion for performing inference with a completely unsupervised modality.

**Student Teacher:** Various knowledge distillation methods use student-teacher models for missing modality inference. They often use an extra auxiliary modality during training to increase a different modality's performance during testing, however, they assume the availability of labeled training data from both modalities (Xue et al., 2022; Kong et al., 2019; Wang et al., 2020; Bruce et al., 2021). Notably, Thoker & Gall (2019) performs knowledge distillation without labels for one modality, i.e. in the UMA setting. However, they require an ensemble of students to perform mutual learning and limit their analysis to one method using two visual modalities on one dataset. Similarly, IMUTube (Kwon et al., 2020) and ChromoSim (Hao et al., 2022) simulate IMU data from videos to teach an IMU model and then perform zero-shot transfer to real IMU data. However, these approaches are resource-intensive, cannot easily extend to other modalities, and face simulation-reality gaps.

**Contrastive Alignment:** Unsupervised contrastive methods, such as IMU and vision-based examples like ImageBind (Girdhar et al., 2023), IMU2CLIP (Moon et al., 2022), and mmg-Ego4d (Gong et al., 2023), have emerged as powerful tools for robust multimodal representation learning and are relevant to our Contrastive Alignment (CA) method. While these approaches have shown promise in alignment for retrieval and generation tasks, they have limitations. ImageBind and IMU2CLIP focus exclusively on egocentric data and IMU2CLIP aligns IMU data with text which violates our UMA setting. Although mmg-Ego4d, addresses UMA in their zero-shot cross-modal transfer task, but their work is limited to a single, private dataset and focuses mainly on few shot label learning. Our CA method addresses these limitations by encompassing both egocentric and third-person public datasets, tackling the UMA setting effectively, and offering flexibility during inference. Uniquely, our approach enables fusion of all available modalities during testing, even when trained on only one labeled modality. We also test for temporal robustness, visualize the latent space, and compare with other UMA methods.

**Cross-modal Transfer Through Time:** Existing video understanding and robust sensor fusion methods often leverage temporal data heavily, employing techniques such as temporal masking and reconstruction (Tong et al., 2022; Kong et al., 2019), spatio-temporal memory banks (Islam et al., 2022), or fusion of temporal chunks through transformer self-attention (Shi et al., 2024; Zhao et al., 2022; Wang et al., 2022b). However, these approaches typically assume the availability of labeled data for all modalities during training, limiting their applicability to the UMA setting. While these methods, inspired by ViT (Arnab et al., 2021) and SwinTransformers (Liu et al., 2022), chunk continuous data to create input tokens for a transformer, they fundamentally differ from our approach. Traditional methods restrict each token to a specific time segment, or shifted time segments, whereas C3T learns time segments through temporal convolutions and performs alignment on these segments across modalities. This novel technique of cross-modal temporal alignment showcases significant potential for further investigation across various research domains.

## 3 BACKGROUND

We investigate the creation of a multi-modal latent space for human action recognition, denoted as $\mathcal{Z}$, that can be leveraged for UMA and exhibits robustness to input variations and noise. In this context, robustness refers to the ability of the latent space to maintain consistent representations despite shifts in input data distribution across modalities or variations in action execution speed or style. We assume that there exists a learnable projection $f^{(k)}$ from every modality $k \in 1, \ldots, M$ to this latent space $\mathcal{Z}$, i.e., $f^{(k)} : \mathcal{X}^{(k)} \rightarrow \mathcal{Z}$, such that the same actions viewed from different modalities map to proximal points in $\mathcal{Z}$, while distinct actions map to disparate regions. We further assume there exists a learnable mapping $h$ from the latent space $\mathcal{Z}$ to the label space of human actions $\mathcal{Y}$, i.e., $h : \mathcal{Z} \rightarrow \mathcal{Y}$. This mapping should be invariant to the originating modality of the latent representation. Our method leverages the intuition that proximity in the high-dimensional space $\mathcal{Z}$ indicates semantic similarity. Points near $z_i \in \mathcal{Z}$ likely map to the same class, allowing classification of neighboring vectors regardless of their originating modality using the same decision boundary $h$. We use cosine similarity to measure "nearness," as it's more effective in high-dimensional spaces than Euclidean distance. This choice: (1) focuses on directional similarity, (2) mitigates the "curse of dimensionality," and (3) aligns with the distributional hypothesis in representation learning. This approach is common in multimodal alignment (Radford et al., 2021).

For simplicity, we experiment with 2 modalities $M = 2$ and assume $n$ data points are split into 4 disjoint index sets $I_1 \cup I_2 \cup I_3 \cup I_4 \in \{1 \ldots n\}$. Under our UMA setting, during training the model has access to 2 of these datasets. One contains labeled data for one modality $\mathcal{D}_{\text{HAR}} = \{(\mathbf{x}_i^1, \mathbf{y}_i)\}_{i=1}^{I_1}$ and the other contains pairs of data between the modalities but these points are unlabeled: $\mathcal{D}_{\text{Align}} = \{(\mathbf{x}_i^1, \mathbf{x}_i^2)\}_{i=1}^{I_2}$. This is analogous to having an existing sensor with labeled data, and introducing a new sensor in which data can be synchronously collected, but there is no additional annotation effort (Figure 1). Given this setup, the key question arises: In the absence of the sensor with labeled data, can the model effectively perform inference using only the unlabeled modality? The third and fourth sets are used for validation and testing and contain only labeled data from the second modality, i.e. $\mathcal{D}_{\text{Val}} = \{(\mathbf{x}_i^2, \mathbf{y}_i)\}_{i=1}^{I_3}$ and $\mathcal{D}_{\text{Test}} = \{(\mathbf{x}_i^2, \mathbf{y}_i)\}_{i=1}^{I_4}$.

## 4 UNSUPERVISED MODALITY ADPATION METHODS

We propose three methods for transferring knowledge to a new sensing modality without exposure to labels in that modality, i.e. Unsupervised Modality Adaptation (UMA): (1) a Student-Teacher approach (ST) (2) a Contrastive Alignment technique (CA), and (3) a method to perform Cross-modal Transfer Through leveraging Temporal information (C3T). In the context of human action or activity recognition (HAR), we conduct experiments using RGB videos as the source domain with some labeled data $x^{(1)} = x^{(\text{RGB})}$ and Inertial Measurement Unit (IMU) data as the target $x^{(2)} = x^{(\text{IMU})}$. Training for UMA occurs in two phases: a) Supervised Learning with RGB data and b) Unsupervised alignment across both modalities, and inference (phase c) occurs on IMU data, as depicted for each method in Figure 2.

**Student Teacher** ST leverages an RGB video model trained in phase a) to produce pseudo-labels to train the IMU model in phase b). In this case, the latent transfer space is the output logit space, $\mathcal{Z} = \mathcal{Y}$, or equivalently the HAR module is the identity $h = \mathbf{1}$, and the cross-modal representations are aligned using the cross-entropy loss $\mathcal{L}_{CE}$, given in Appendix Equation (2). We denote the teacher network as $f^{(1)} : \mathcal{X}^{(1)} \rightarrow \mathcal{Y}$ and the student network as $f^{(2)} : \mathcal{X}^{(2)} \rightarrow \mathcal{Y}$. First, we train the teacher $f^{(1)}$ on $\mathcal{D}_{\text{HAR}}$ with labeled RGB data. Next, in order to train $f^{(2)}$ on $\mathcal{D}_{\text{Align}}$, which does not contain labels, we first use $f^{(1)}(x_i^{(1)}) = \hat{y}_i^{(1)}$ to generate pseudo-labels for every datapoint $i \in I_2$. Then we use the augmented dataset $\hat{\mathcal{D}}_{\text{Align}} = \{(\mathbf{x}_i^1, \mathbf{x}_i^2, \hat{\mathbf{y}}_i^{(1)})\}_{i=1}^{I_2}$ to train $f^{(1)}$. The teacher network minimizes $\mathcal{L}_{CE}(P_{f^1(x)}, P_y)$ and the student minimizes $\mathcal{L}_{CE}(P_{f^2(x)}, P_{f^2(x)})$, where $P$ represents the ground truth distribution or the distribution given by the model's outputs logits.

**Contrastive Alignment** In our work, CA performs phase a) Supervised RGB training in the same fashion as the student teacher, however, it uses an a model with two parts: An encoder $f^{(1)}$ to extract the latent variable $z$, and a task specific MLP head $h$. The extracted latent space $\mathcal{Z}$ allows for scalability and interoperability of adding different sensing modalities, types of encoders, and output task heads.

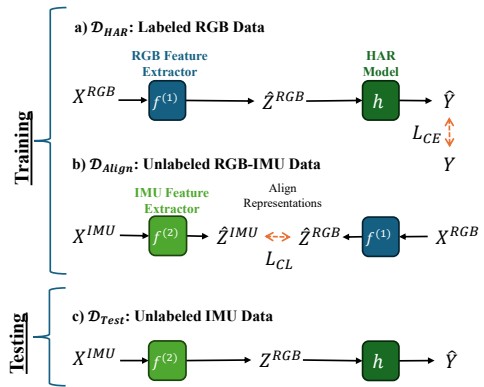

(i) **Student Teacher (ST):** Trains a model with the teacher modality in a) and uses it to psuedolabel the data in b) and train the student model.

(ii) **Contrastive Alignment (CA):** Trains a feature extractor and task module in a), and aligns each modality's feature extractor in b).

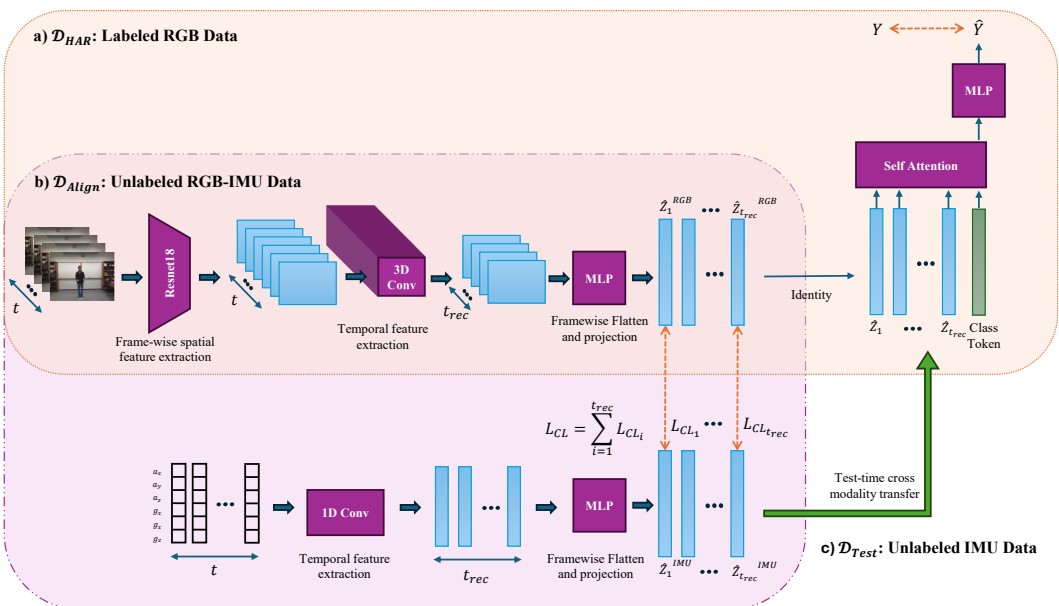

(iii) **Cross-modal Transfer Through Time (C3T):** Is similar to CA, however, it aligns representations across the temporal receptive field of a 3D convolution and uses self-attention across this latent time sequence to perform HAR inference.

Figure 2: **Training and Testing in the UMA setting:** Training happens in two phases: a) trains the HAR model on labeled RGB inputs and b) aligns unlabeled IMU and RGB modalities. UMA testing c) occurs on unlabled IMU data.

In phase b) of training, CA performs unsupervised contrastive alignment with the outputs of the the RGB encoder $f^{(1)}$ and the IMU encoder $f^{(2)}$ on unlabeled data. To align different modalities in the feature space on $\mathcal{D}_{\text{Align}}$ we use a symmetric contrastive loss formulation $\mathcal{L}_{CL}$ (Radford et al., 2021; Moon et al., 2022; Girdhar et al., 2023; Gong et al., 2023) with temperature parameter $\tau$:

$$\mathcal{L}_{CL} = -\frac{1}{N}\sum_{i=1}^{N}\log\frac{\exp(\langle \hat{z}_i^{(1)}, \hat{z}_i^{(2)}\rangle/\tau)}{\sum_{i=1}^{N}\exp(\langle \hat{z}_i^{(1)}, \hat{z}_i^{(2)}\rangle/\tau)}, \text{ where } \hat{z}_i^{(k)} = \frac{f^{(k)}(x_i^{(k)})}{||f^{(k)}(x_i^{(k)})||}, k \in \{1,2\} \quad (1)$$

The symmetric contrastive loss will cluster representations in $\mathcal{Z}$ by cosine similarity, which brings about the desired property of the latent space that vectors of the same class are near each other.

**Cross-modal Transfer Through Time** CA and ST do not leverage latent temporal information as they collapse the entire time sequence into one latent vector. We thus propose a Cross-modal Transfer Through Time (C3T) model that leverages the temporal information of time-series sensors when aligning and fusing their representations. C3T removes the MLP layer at the end of the feature

Table 1: **UMA vs. Supervised Performance:** The modules $f^{(1)}$, $f^{(2)}$, and $h$ can operate in supervised or UMA (ST, CA, CT3) modes. Top-1 and Top-3 accuracies shown.

|  | Model | UTD-MHAD Top-1 | UTD-MHAD Top-3 | CZU-MHAD Top-1 | CZU-MHAD Top-3 | MMACT Top-1 | MMACT Top-3 | MMEA-CL Top-1 | MMEA-CL Top-3 |
|---|---|---|---|---|---|---|---|---|---|
| Supervised | IMU | **87.9** | **97.7** | **95.1** | 98.2 | 70.0 | 90.0 | 65.8 | 87.6 |
| | RGB | 53.8 | 73.1 | 94.0 | **99.7** | 42.1 | 61.6 | 54.2 | 77.1 |
| | Fusion | 62.5 | 82.2 | 95.0 | 98.5 | **76.7** | **92.0** | **80.1** | **92.7** |
| UMA | Random | 3.7 | 11.1 | 4.6 | 16.6 | 2.9 | 8.6 | 3.1 | 9.4 |
| | ST | 12.9 | 24.6 | 41.1 | 61.9 | 17.6 | 34.7 | 9.9 | 22.7 |
| | CA | 42.6 | 67.4 | 70.0 | 92.7 | 24.5 | 47.6 | 29.3 | 51.7 |
| | C3T | **62.5** | **86.4** | **84.2** | **96.7** | **32.4** | **57.9** | **51.2** | **78.8** |

Table 2: **Data Splits for Unsupervised Modality Adaptation (UMA)**: During training, no labeled IMU data is present, thus the model can only leverage the correlations between $X^{\text{IMU}}$ and $X^{\text{RGB}}$ to learn classes.

| Split | $X^{\text{RGB}}$ | $X^{\text{IMU}}$ | $Y$ | % of Data |
|---|---|---|---|---|
| $\mathcal{D}_{\text{HAR}}$: Train a) | ✓ | | ✓ | 40% |
| $\mathcal{D}_{\text{Align}}$: Train b) | ✓ | ✓ | | 40% |
| $\mathcal{D}_{\text{Val}}$: Val | | ✓ | ✓ | 10% |
| $\mathcal{D}_{\text{Test}}$: Test | | ✓ | ✓ | 10% |

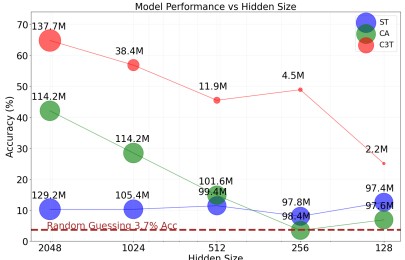

Figure 3: **Performance vs Model Size:** Bubbles show accuracy for each latent space hidden size. Bubble size indicates model parameters (millions).

encoders and uses the output map of the temporal convolutions directly. This approach leverages the temporal receptive field, which represents the span of input time steps that influence each output feature vector. We use this set of output vectors $z_t$ as latent vectors for $t = 1 \ldots t_{\text{rec}}$ where $t_{\text{rec}}$ is the length of the receptive field. Then during the alignment phase, we align each of these time vectors with the same time vector from the other modality. Each of the modality feature encoder convolutional structures was carefully designed to result in the same number of output feature vectors. When training the HAR model, we use self attention with a learned class token to predict the action. The intuition is that the encoder will learn which tokens over time are the most informative for the action class and predict accordingly. This is a common method to perform classification with transformers (Dosovitskiy et al., 2020; Carion et al., 2020; Meinhardt et al., 2022). Our implemented HAR task head is most notably similar to the ViT architecture (Dosovitskiy et al., 2020), but instead of inputs being image chunks, they are temporal feature chunks.

## 5 EXPERIMENTAL RESULTS

**Implementation:** Our methods utilize three key neural network modules:

*(1) Video Feature Encoder* $f^{(1)} : \mathcal{X}^{(1)} \to \mathcal{Z}$: We employ a pretrained ResNet18 on every frame of the video followed by 3D convolution and a 2-layer MLP with ReLU activations. The Resnet is a well-established lightweight spatial feature extractor (Hara et al., 2017), and the 3D convolution is an effective temporal feature extractor (Tran et al., 2018).

*(2) IMU Feature Encoder* $f^{(2)} : \mathcal{X}^{(2)} \to \mathcal{Z}$: A 1D CNN followed by an MLP is used here. CNNs have shown superior performance in extracting features from time-series data like IMU signals, efficiently capturing local patterns and temporal dependencies (Valarezo et al., 2017). Both feature encoders output the label for ST, the latent dimension vector for CA, and $t_{\text{rec}}$ latent vectors for C3T.

*(3) HAR Task Decoder* $h : \mathcal{Z} \to \mathcal{Y}$: ST does not require this module and CA uses a simple MLP. C3T employs a self-attention module to better capture long-range dependencies in the latent space, which is particularly beneficial for complex action sequences (Moutik et al., 2023).

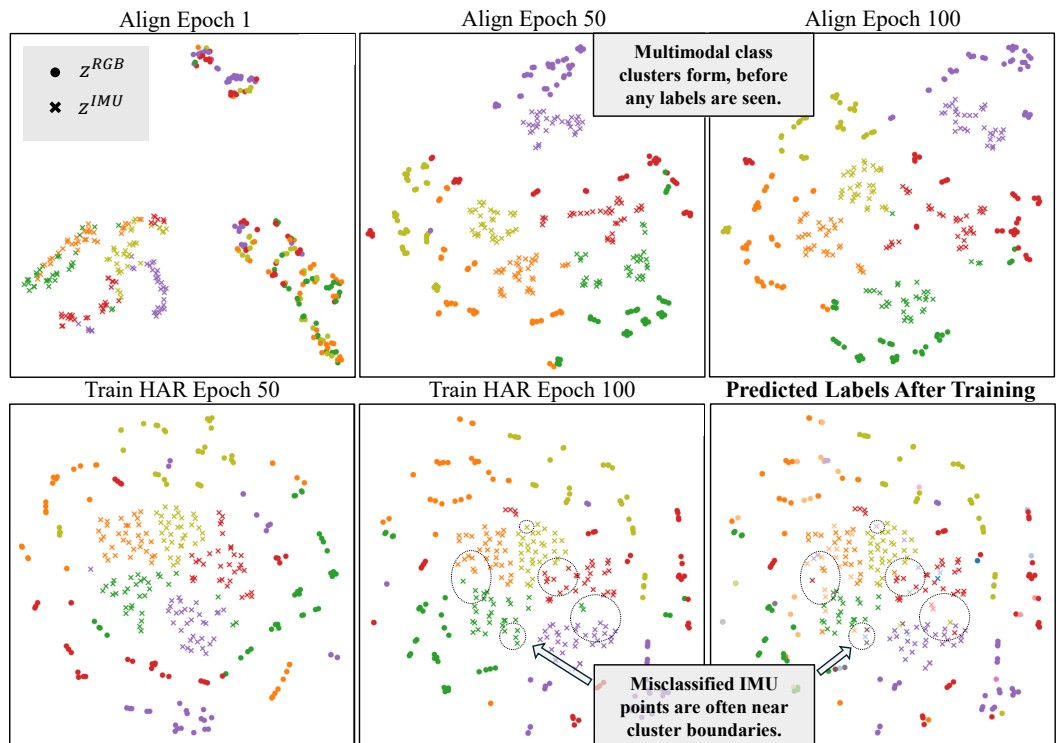

Figure 4: **CA TSNE Plots in UMA Training Method 1:** The following shows the progression of the latent representations of datapoints for 5 classes (Bowling, Clap, Draw circle (clockwise), Jog, Basketball shoot) during training CA on the UTD-MHAD dataset. At the end we plot the predicted labels and circle areas of confusion, which seems to often occur at the boundaries between clusters.

Our simplified models with comparable module sizes across methods isolate the impact of our novel technique, demonstrating its efficacy independent of complex architectures or larger datasets, while allowing for potential future scaling and expansion.

**Datasets:** We present results on four diverse datasets: (1) UTD-MHAD (Chen et al., 2015), a small yet structured dataset; (2) CZU-MHAD (Chao et al., 2022), a slightly larger dataset captured in a controlled environment; (3) MMACT (Kong et al., 2019), a very large dataset with various challenges; and (4) MMEA-CL (Xu et al., 2023), an egocentric camera dataset. For each dataset, we create an approximately 40-40-10-10 percent data split for the $\mathcal{D}_{\text{Align}}$, $\mathcal{D}_{\text{HAR}}$, $\mathcal{D}_{\text{Val}}$, and $\mathcal{D}_{\text{Test}}$ sets respectively, as shown in Table 2. $\mathcal{D}_{\text{Val}}$ was used to perform a minor hyperparameter search on the UTD-MHAD dataset. The methods performed best with a learning rate of $1.5 \times 10^{-4}$, a batch size of 16, and a latent representation dimension of 2048 with an Adam optimizer. Experiments were implemented in Pytorch and run on a 16GB NVIDIA Quadro RTX 5000, four 48GB A40s, or four 48GB A100s. More detailed information about each dataset and implementation can be found in Appendix A.4.

**Quantitative Results:** Our results show our methods' Top-1 and Top-3 test accuracies on 4 different datasets (Table 1). Our modular architecture also adapts to the supervised setting: IMU trains $f^{(1)} : \mathcal{X}^{(\text{IMU})} \rightarrow \mathcal{Y}$, RGB trains $f^{(2)} : \mathcal{X}^{(\text{RGB})} \rightarrow \mathcal{Y}$, and fusion averages the outputs of $f^{(1)}$ and $f^{(2)}$ with a linear head $h$. The supervised setting serves as a strong empirical upper bound for our methods to approach, thus we use it as a comparison. We ran each experiment thrice with different random seeds, reporting the average accuracies, to ensure rigorous empirical results. Full trial data and standard deviations are available in Appendix Figure 6.

The performance rankings of our UMA methods were consistent, with C3T consistently achieving the highest accuracy, followed by CA, while ST consistently ranked the lowest. We believe ST performs the worst since the student is bound by the performance of the teacher. Thus, given the noise in the teacher distribution, the student also suffers in performance. Furthermore, given that the latent space is the label distribution, ST provides little flexibility in extending to various modalities and

Table 3: **Additional Experiments**: Performance of ST, CA, and C3T across various training methods, modalities, and noise. All results report UMA accuracy on IMU data, except modality test 2. and 3.

| Model | Training | | | | Modality Testing | | | Noise Experiments | | | | |
|---|---|---|---|---|---|---|---|---|---|---|---|---|
| | 1. Align | 2. HAR | 3. Inter | 4. Comb | 1. IMU | 2. RGB | 3. Both | 1. Crop | 2. Misalign | 3. Dilate | 4.All | None |
| ST | - | - | - | - | 12.9 | 53.8 | 17.0 | 3.4 | 5.7 | 5.7 | 10.2 | 12.9 |
| CA | 38.6 | 43.2 | 27.3 | **42.6** | 42.6 | 56.8 | 60.2 | 10.2 | 2.3 | 21.6 | 18.2 | 42.6 |
| C3T | **62.5** | 35.2 | 51.1 | 27.9 | **62.5** | **78.4** | **79.5** | **52.3** | **46.6** | **56.8** | **58.0** | 62.5 |

tasks. Nonetheless, ST demonstrates a 3-10x improvement over random guessing, indicating some effectiveness in UMA. Unlike CA and C3T, whose performance declines with smaller hidden sizes (Figure 3), ST maintains consistent performance. This difference arises because, in CA and C3T, the hidden size directly influences the latent vector $z$, impacting overall model size. In contrast, ST's hidden size primarily affects the internal latent space within $f^{(2)}$ between convolutional outputs and the MLP classification head. Notably, ST begins to outperform CA at smaller hidden sizes, while C3T retains superior performance even with a reduced model size. This highlights C3T's efficiency and suitability for resource-constrained environments, such as on-device computing for wearables or smart devices, where effective learning of new modalities is essential.

The most notable quantitative result, is that on a few datasets *CA and C3T surpass the supervised RGB model*. We believe that this is because *these methods leverage an inherent correlation between the modalities*, and for some of these datasets the IMU data is more informative than the RGB data. The superiority of the IMU modality in our test scenarios is further evidenced by the supervised unimodal IMU model outperforming both the RGB and fusion models.

**Qualitative Results:** We visualized the latent space outputs of the CA model using TSNE plots (Figure 4). These plots show training when the alignment phase (phase b) is performed first, and then labeled-RGB training (phase a) is performed. The model quickly segments classes during the align phase, even without labels, suggesting that the data's natural structure facilitates class distinction across different modalities. This implies that our methods could potentially adapt to new class labels during testing with just a few samples, as the latent structure would have already grouped similar classes. Furthermore, after alignment and HAR training, we notice how the model tends to misclassify points that are near the boundary between clusters. These visualizations support our initial hypothesis (Figure 1) on how a joint latent space could be leveraged to effectively perform UMA, by using a classification head trained only on RGB data.

Interestingly, we observed that IMU data points consistently cluster towards the center of the plot, with RGB points surrounding them (more examples in Appendix A.6). This pattern persists even in early alignment stages, suggesting it's not solely due to labeled RGB HAR training. While this might indicate that RGB data is more informative, it contradicts our quantitative findings where supervised IMU models outperform RGB models for our given datasets. This phenomenon warrants further investigation as it may have implications for continual learning, test-time adaptation, or domain adaptation, where different modalities should be leveraged differently depending on their placement in the shared latent space. TSNE plots for C3T are provided in Appendix A.6 but were excluded from this discussion due to their difficulty to interpret.

### 5.1 ADDITIONAL EXPERIMENTS:

We conduct three primary sets of experiments to further gain insights from our UMA methods: one investigates various training mechanisms of CA and C3T, the second examines the deployment of each model in different testing scenarios, and the third tests performance under temporal noise.

***How do we train the CA and C3T Architectures?***

**Experimental setup:** We experimented with four ways to perform the two phases of training for CA and C3T. In this setup, all tests were done on the IMU data in $\mathcal{D}_{Test}$.

1. **Align First:** First align the representations generated by the RGB and IMU encoders, $f^{(1)}$ and $f^{(2)}$ RGB on $\mathcal{D}_{Align}$ (phase b depicted in Figure 2). Then freeze the weights for both encoders and train the HAR module $h$ on RGB data in $\mathcal{D}_{HAR}$ (phase a in Figure 2).
2. **HAR First:** Reverse the order from the previous method, first perform phase a), RGB HAR training, then freeze the RGB encoder $f^{(1)}$ and align the IMU encoder $f^{(2)}$, phase b).

3. **Interspersed Training:** Intermittently learn from $\mathcal{D}_{\text{Align}}$ and $\mathcal{D}_{\text{HAR}}$. The model learns an epoch from a) and updates its weights to train the RGB HAR model, then learns an epoch from b) and updates it's weights to align the encoders. The model continues to iterate between the two losses.

4. **Combined Loss:** Train both phases a) and b) but within the same loss iteration. The loss from a) on a batch of data from $\mathcal{D}_{\text{HAR}}$ is added to the loss from b) on a batch of data from $\mathcal{D}_{\text{Align}}$ and the total loss is then used to update the weights of the model: $\mathcal{L}_{\text{Total}} = \mathcal{L}_{\text{CE}} + \mathcal{L}_{\text{CL}}$.

**Results:** As shown in Table 3 **Training**, method 1. Align First performs the best for C3T, whereas 4. Combined Loss performed the best for CA. The main experiments reported in this work (Table 1 use training methods 1. and 4. for C3T and CA, respectively. We hypothesize that in method 2. training the HAR model first yields a latent space that captures the best HAR features for RGB data, which is not directly applicable to IMU data. This implies that cross-modal alignment may be a more difficult task, i.e. adding a modality may require more restructuring of the latent space than adding an output head. This reaffirms findings from existing methods such as CLIP (Radford et al., 2021), where linear probing is used on aligned representations to learn new tasks. Method 3. faced instability in training and was unable to converge. In method 4, for CA we believe one loss acted as a regularizer for the other pushing the latent space $\mathcal{Z}$ to the ideal balance for cross-modal transfer in HAR. However, given that C3T is a more difficult alignment problem, the contrastive loss was dominating the overall loss term preventing the model from learning to perform HAR well.

*Can UMA methods retain performance on the labeled modality? Can they leverage both modalities?*

**Experimental setup:** Table 3**Modality Testing** shows the result of training in the UMA setting, but testing with all combinations of the modalities. Any inputs can be used to perform HAR by simply using the HAR module $h$ on an estimate for the latent vector, $\hat{z}$ derived from the modalities. For ST, $h$ can be viewed as the identity, and $z$'s are the output logits.

1. **RGB (Supervised Learning):** Tests the model on different samples of the training distribution, which is RGB data for our setup. Thus the estimated latent vector is given by $f^{(1)}(x_i^{(1)}) = \hat{z}_i$.

2. **IMU (Cross-modal Zero-Shot Transfer):** Is the main result of this paper and described above in Section 4. Here the estimated latent vector is given by $f^{(2)}(x_i^{(2)}) = \hat{z}_i$.

3. **Both (Sensor Fusion):** Merges latent vectors from each modality. Assuming each estimate is equally as good as the other: $\hat{z}_i = \mathbb{E}[z_i | x_i^{(1)}, x_i^{(2)}] = E[z_i | \hat{z}_i^{(1)}, \hat{z}_i^{(2)}] = \frac{\hat{z}_i^{(1)} + \hat{z}_i^{(2)}}{|\hat{z}_i^{(1)} + \hat{z}_i^{(2)}|}$ . Given that we align the latent vectors from different modalities by minimizing the angle between them, i.e. cosine similarity, we also fuse vectors by generating the normalized vector whose angle is halfway between the estimated vectors. This is a relatively unexplored fusion approach as most sensor fusion methods sum, average, concatenate, use attention, or apply a learned module over features.

**Results:** As expected, Table 3 shows that C3T outperforms the other methods. When comparing the performances in the different test scenarios, intriguingly our experiments indicate that when given both modalities, fusion performs better than the RGB model alone. Instead of introducing noise or uncertainty into the model, introducing an unlabeled modality may add structure to the shared latent space that bolsters performance, especially if that modality is highly informative for the given task. This observation bears resemblance to knowledge distillation methods, where an auxiliary modality during training leads to improved testing outcomes, however, these methods usually assume that auxiliary modality is labeled (Chen et al., 2023). We believe using UMA for sensor fusion is an interesting direction that should be explored further in future works.

*How robust are these methods to time-related noise?*

**Experimental Setup:** We evaluated each method's robustness to temporal noise during testing on $\mathcal{D}_{\text{test}}$, simulating three real-life scenarios (Table 3):

1. **Crop:** Randomly shifts and crops both modalities' time sequences by 60%, simulating continuous real-time action recognition, where an action may not occur in the middle of the time sequence.

2. **Misalign:** Applies crop to one modality, mimicking hardware asynchrony or differing framerates.

3. **Dilation:** Applies crop to both modalities and then upsamples, imitating slower action movements.

**Results:** C3T demonstrates robust performance under temporal noise, likely due to its attention-based HAR module leveraging tokens from the receptive field of a temporal convolution. The self-attention mechanism compares neighboring tokens from various time sections, effectively capturing actions

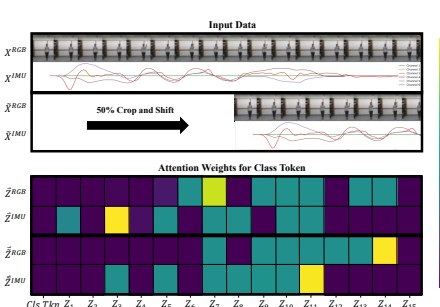

Figure 5: **Attention heatmap for C3T's HAR module:** Input shifts shift the attention weights of the temporal latent vectors.

Table 4: **C3T HAR Module Ablations:** Comparison of 2 Attention methods and two MLP methods.

| Input | Attention | | MLP | |
|---|---|---|---|---|
| | Class | Concat | Add | Concat |
| Clean | 62.5 | 44.32 | 56.82 | **70.45** |
| Noisy | **52.27** | 43.18 | 50.00 | 43.18 |

Table 5: **Architecture Ablation:** Comparison of different architectures for RGB and IMU encoders across methods. Encoder types are reported as (RGB-Spatial / RGB-Temporal / IMU-Temporal), where C = Convolutional and A = Attention.

| Method | Encoders | Params (M) | Acc. (%) |
|---|---|---|---|
| ST | C/C/C | 129.2 | **12.9** |
| | C/C/A | 97.8 | 10.2 |
| | C/A/C | 871.2 | 11.4 |
| | A/C/C | 291.5 | 5.7 |
| CA | C/C/C | 163.8 | **38.6** |
| | C/C/A | 132.3 | 19.3 |
| | C/A/C | 905.7 | 34.1 |
| | A/C/C | 326.0 | 26.1 |
| C3T | C/C/C | 137.7 | **62.5** |
| | C/C/A | 106.3 | 15.9 |
| | C/A/C | 879.6 | 53.4 |
| | A/C/C | 300.0 | 33.0 |

regardless of their temporal position within the sequence. Visually, we observe the attention weights over the latent vectors in C3T, and can see a shift when we shift the input sample (Figure 5). Additionally, C3T's attention head's ability to handle variable-length inputs during inference provides an advantage in cropped scenarios, requiring minimal zero-padding compared to ST and CA methods.

## 5.2 ABLATIONS:

We conducted comprehensive ablation experiments on our model architecture (Table 5), comparing convolutional and attention modules for RGB (spatial and temporal) and IMU (temporal) encoders. Results generally favored convolutional architectures across various methods in our UMA setting. Notably, C3T's superior performance cannot be attributed solely to its attention head leveraging temporal information, as ST or CA models with temporal attention did not perform comparably well. Instead, C3T's effectiveness stems from its unique method of alignment in the temporal space.

Further ablation on C3T head architectures (Table 4) compared the class token-based self-attention head with alternatives: concatenating output attention tokens and projecting, adding latent vectors $Z_1 \ldots Z_{t_{rec}}$ and passing through an MLP, and concatenating vectors and using an MLP. The latter two methods do not use attention. While concatenating latent vectors and using an MLP performed best on clean data, the class token attention mechanism offered superior robustness to noise. The attention visualization in Figure 5 corroborates these findings, showing the class token approach's resilience to shift noise. In addition, we notice all C3T variants outperformed CA and ST in UMA performance (Table 1) on the UTD-MHAD dataset, emphasizing C3T's strength in temporal alignment, regardless of the classification head.

## 6 CONCLUSION

We explored the UMA framework for human action recognition, challenging models to perform inference with an unlabeled modality during training. Our experiments focused on constructing a unified latent space between modalities and comparing three UMA methods in various settings. Our C3T method, integrating temporal convolutions with self-attention, showed promising results for robust cross-modal transfer in UMA. Future work could explore continuous HAR with temporal localization, and generalization to additional modalities and tasks. We hope our results inspire further exploitation of cross-modal latent spaces for more robust human motion understanding in AI models.

## 7 REPRODUCABILITY STATEMENT

We have made significant efforts to ensure the reproducibility of our work. Our commitment to reproducibility extends to several key areas:

1. Code Availability: The complete codebase for this work, including all models, training scripts, and evaluation procedures, is provided in the supplementary materials. Upon acceptance, this code will be open-sourced and made publicly available on GitHub.

2. Dataset Preparation: Detailed instructions for dataset setup, including any preprocessing steps and data splits used in our experiments, are provided in the supplementary materials Appendix A.4. This ensures that other researchers can replicate our exact experimental conditions.

3. Hardware and Hyperparameters: A comprehensive description hyperparameters used in our experiments is provided in Section 5. This includes model-specific parameters, optimization settings, the GPUs used for all experiments, and any other configuration details necessary for reproduction.

4. Architecture Transparency: We are very transparent about our model architectures in Section 5, ensuring others can understand our code our reconstruct such models themselves.

5. Random Seed Control: As mentioned, all key experiments were conducted with three different random seeds. The specific seeds used are documented in the code as well as Appendix A, allowing for exact replication of our experimental conditions. The full results across all trials is given in Table 1.

Evaluation Metrics: The precise definitions of all evaluation metrics used in this study are provided in Section 4 of the main paper, with additional details in Appendix D. By providing this comprehensive set of resources and information, we aim to facilitate the reproduction of our results by the research community. We believe that this level of transparency is crucial for advancing the field and allowing for thorough validation and extension of our work.

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

# A APPENDIX: MAIN EXPERIMENTS

Below we report some addition Unsupervised Modality Adaptation (UMA) experiments. All tables report the accuracy on the $\mathcal{D}_{Test}$ for each method ( accuracy $= \frac{1}{I_4} \sum_{i=1}^{I_4} \mathbb{1}_{\hat{\mathbf{y}}_i = \mathbf{y}_i}$). In particular, we attempt to implement baselines as described in the next section, however, all existing methods perform poorly.

For reproducibility and to guarantee scientific rigour of our experiments, our main results table in Table 1 ran 3 times with pytorch random seeds 1,2, and 3. The full results are given in Figure 6. We also note that the supervised methods were only trained on 40% of the data given in $\mathcal{D}_{\text{HAR}}$ for a fair comparison to the other methods.

## A.1 FULL RESULTS:

## A.2 METHODS:

**Student Teacher:**

$$\mathcal{L}_{CE}(P_{\hat{y}}, P_y) = -\frac{1}{N} \sum_{i=1}^{N} \sum_{j=1}^{C} \mathbb{1}_{y_i=j} \log\left(\frac{\exp \hat{y}_{i,j}}{\sum_{i=1}^{M} \exp \hat{y}_{i,j}}\right) \tag{2}$$

where $\hat{y}_i$ is the output of the $i$th sample in the batch of $N$ samples, $\hat{y}_{i,j}$ is the score for the $j$th class out of $C$ classes, and $P_y$ represents the probability distribution produced by a given model's output logits. The teacher network minimizes $\mathcal{L}_{CE}(P_{f^1(x)}, P_y)$ and the student minimizes $\mathcal{L}_{CE}(P_{f^2(x)}, P_{f^2(x)})$. Since the student approximates the teacher and the teacher approximates the true distribution, this implies that the student can only be as good as the teacher at approximating the true distribution:

$$\mathcal{L}_{CE}(P_{f^1(x)}, P_y) \leq \mathcal{L}_{CE}(P_{f^2(x)}, P_y) \tag{3}$$

## A.3 C3T MODULES

Here we provide a more precise formulation of the modules used for C3T.

The updated modules are as follows:
**Video Feature Encoder** $f^{(1)} : \mathcal{X}^{(1)} \to \mathcal{Z}^{t_{rec}}$: This module applies a pretrained Resnet18 to every frame a video and then performs a single 3D convolution. The resulting output is a set $t_{rec}$ $z$: $\hat{\mathbf{Z}}^{(1)} = (\hat{z}_1^{(1)} \dots \hat{z}_{t_{rec}}^{(1)})$.
**IMU Feature Encoder** $f^{(2)} : \mathcal{X}^{(2)} \to \mathcal{Z}^{t_{rec}}$: This is a 1D CNN that decreases the time dimension to $t_{rec}$, resulting in an output of $\hat{\mathbf{Z}}^{(2)} = (\hat{z}_1^{(2)} \dots \hat{z}_{t_{rec}}^{(2)})$.
**HAR Task Decoder** $h : \mathcal{Z}^{t_{rec}} \to \mathcal{Y}$: This module is like a transformer encoder that uses self-attention on an input sequence of length $t_{rec}$ vectors appended with a learned class token. The output class token of the self attention layer is then passed through a FFN and outputs a single action label $y_i$.

## A.4 DATASETS

Here we provide more information on the datasets and how they were used in our experiments.

**UTD-MHAD** Most of the development and experiments were performed on the UTD-Multi-modal Human Action Dataset (UTD-MHAD) (Chen et al., 2015). This dataset consists of roughly 861 sequences of RGB, skeletal, depth and an inertial sensor, with 27 different labeled action classes performed by 8 subjects 4 times. The inertial sensor provided 3-axis acceleration and 3-axis gyroscopic information, and all 6 channels were used for in our model as the IMU input. Given our motivation, we only use the video and inertial data; however, CA can easily be extended to multiple modalities.

**CZU-MHAD** The Changzhhou MHAD (Chao et al., 2022) dataset provides about 1,170 sequences and includes depth information from a Kinect camera synchronized with 10 IMU sensors, each 6 channels, in a very controlled setting with a user directly facing the camera for 22 actions. They do not provide RGB information, thus we use depth as the visual modality, broadcast it to 3 channels,

| | | | UTD | | | | MMACT | | | | MMEA | | | | CZU | | | |
|---|---|---|---|---|---|---|---|---|---|---|---|---|---|---|---|---|---|---|
| | | | Top1 | Top3 | Top5 | Top7 | Top1 | Top3 | Top5 | Top7 | Top1 | Top3 | Top5 | Top7 | Top1 | Top3 | Top5 | Top7 |
| Supervised | RGB | trial 1 | 61.36364 | 75 | 79.54546 | 82.95454 | 42.37089 | 63.26291 | 74.53052 | 82.39436 | 56.70732 | 79.87805 | 86.28049 | 91.46342 | 94.64286 | 100 | 100 | 100 |
| | | trial 2 | 61.36364 | 79.54546 | 86.36364 | 90.90909 | 41.19718 | 59.50704 | 69.71831 | 79.10798 | 56.09756 | 78.5061 | 85.97561 | 89.93903 | 96.42857 | 100 | 100 | 100 |
| | | trial 3 | 38.63636 | 64.77273 | 67.04546 | 72.72727 | 42.60563 | 62.0892 | 72.1831 | 78.05164 | 49.84756 | 72.86585 | 80.18293 | 85.97561 | 91.07143 | 99.10714 | 99.10714 | 100 |
| | | avg | 53.78788 | 73.10606 | 77.65152 | 82.19697 | 42.0579 | 61.61972 | 72.14398 | 79.85133 | 54.21748 | 77.08333 | 84.14634 | 89.12602 | 94.04762 | 99.70238 | 99.70238 | 100 |
| | | std | 13.1216 | 7.566283 | 9.797361 | 9.114553 | 0.754589 | 1.921445 | 2.406344 | 2.26478 | 3.796723 | 3.716307 | 3.4358 | 2.8328 | 2.727723 | 0.515493 | 0.515493 | 0 |
| | IMU | trial 1 | 87.5 | 97.72727 | 97.72727 | 97.72727 | 69.24883 | 90.14085 | 95.30516 | 96.83099 | 65.70122 | 87.5 | 92.68293 | 96.18903 | 94.64286 | 98.21429 | 98.21429 | 100 |
| | | trial 2 | 87.5 | 97.72727 | 97.72727 | 97.72727 | 70.30516 | 90.14085 | 94.95305 | 96.71362 | 66.46342 | 87.80488 | 93.14024 | 95.27439 | 95.53571 | 98.21429 | 98.21429 | 100 |
| | | trial 3 | 88.63636 | 97.72727 | 97.72727 | 97.72727 | 70.30516 | 89.78873 | 94.95305 | 96.47887 | 65.2439 | 87.5 | 92.53049 | 95.42683 | 94.64286 | 98.21429 | 100 | 100 |
| | | avg | 87.87879 | 97.72727 | 97.72727 | 97.72727 | 69.95305 | 90.02348 | 95.07042 | 96.67449 | 65.80285 | 87.60163 | 92.78455 | 95.63008 | 94.94048 | 98.21429 | 98.80953 | 100 |
| | | std | 0.656078 | 1.74E-14 | 1.74E-14 | 1.74E-14 | 0.609872 | 0.203297 | 0.203291 | 0.179291 | 0.616079 | 0.176023 | 0.317324 | 0.490026 | 0.515487 | 0 | 1.03098 | 0 |
| | FUSION | trial 1 | 64.77273 | 87.5 | 92.04546 | 94.31818 | 77.34742 | 92.13615 | 94.71831 | 96.12676 | 83.84146 | 94.96951 | 96.95122 | 97.40854 | 95.53571 | 98.21429 | 99.10714 | 100 |
| | | trial 2 | 51.13636 | 76.13636 | 81.81818 | 86.36364 | 75.93896 | 90.96244 | 95.89202 | 97.65258 | 71.18903 | 88.26219 | 92.22561 | 94.05488 | 93.75 | 99.10714 | 100 | 100 |
| | | trial 3 | 71.59091 | 92.95454 | 88.63636 | 89.77273 | 76.87794 | 92.95775 | 95.53991 | 96.3615 | 85.21342 | 94.81707 | 96.79878 | 97.40854 | 95.53571 | 98.21429 | 99.10714 | 99.10714 |
| | | avg | 62.5 | 82.19697 | 87.5 | 90.15152 | 76.72144 | 92.01878 | 95.38341 | 96.71361 | 80.0813 | 92.68292 | 96.29065 | 96.29065 | 94.94047 | 98.51191 | 99.40476 | 99.70238 |
| | | std | 10.41495 | 5.719573 | 5.207475 | 3.990775 | 0.717153 | 1.00282 | 0.602302 | 0.821596 | 7.731427 | 3.829226 | 2.685408 | 1.936237 | 1.03098 | 0.515487 | 0.515493 | 0.515493 |

| | | | UTD | | | | MMACT | | | | MMEA | | | | CZU | | | |
|---|---|---|---|---|---|---|---|---|---|---|---|---|---|---|---|---|---|---|
| | | | Top1 | Top3 | Top5 | Top7 | Top1 | Top3 | Top5 | Top7 | Top1 | Top3 | Top5 | Top7 | Top1 | Top3 | Top5 | Top7 |
| UMA | ST | trial 1 | 15.90909 | 27.27273 | 32.95454 | 37.5 | 17.60563 | 37.0892 | 48.94366 | 56.69014 | 9.60366 | 23.17073 | 30.79268 | 38.71951 | 40.17857 | 59.82143 | 70.53571 | 79.46429 |
| | | trial 2 | 13.63636 | 23.86364 | 30.68182 | 30.68182 | 18.5446 | 36.2676 | 47.30047 | 55.86855 | 9.29878 | 20.73171 | 27.7439 | 37.19512 | 41.07143 | 64.28571 | 69.64286 | 74.10714 |
| | | trial 3 | 9.09091 | 22.72727 | 30.68182 | 35.22727 | 16.66667 | 30.86855 | 38.7324 | 46.94836 | 10.82317 | 24.2378 | 34.7561 | 43.44512 | 41.96429 | 61.60714 | 68.75 | 77.67857 |
| | | avg | 12.87879 | 24.62121 | 31.43939 | 34.4697 | 17.60563 | 34.74178 | 44.99218 | 53.16902 | 9.908537 | 22.71341 | 31.09756 | 39.78658 | 41.07143 | 61.90476 | 69.64286 | 77.08333 |
| | | std | 3.471647 | 2.365532 | 1.312156 | 3.471647 | 0.938965 | 3.37938 | 5.48303 | 5.402886 | 0.806631 | 1.797226 | 3.516028 | 3.258774 | 0.89286 | 2.246972 | 0.892855 | 2.727727 |
| | CA | trial 1 | 40.69767 | 79.06977 | 79.06977 | 84.88372 | 23.7927 | 47.34982 | 59.31732 | 73.38045 | 29.09648 | 50.38285 | 61.8683 | 69.67841 | 66.36364 | 94.54546 | 98.18182 | 100 |
| | | trial 2 | 38.37209 | 60.46511 | 70.93023 | 80.23256 | 27.79741 | 52.06125 | 65.01767 | 75.02945 | 30.62787 | 55.43845 | 65.3905 | 71.97549 | 79.0091 | 94.54546 | 94.54546 | 100 |
| | | trial 3 | 48.83721 | 75.5814 | 86.04651 | 93.02325 | 21.90813 | 43.4629 | 59.95288 | 68.55124 | 28.17764 | 49.31087 | 62.17458 | 71.05686 | 64.54546 | 89.09091 | 94.54546 | 96.36364 |
| | | avg | 42.63566 | 67.44186 | 78.68217 | 86.04651 | 24.49941 | 47.62466 | 61.56262 | 72.32038 | 29.30066 | 51.71006 | 63.14446 | 70.90352 | 70 | 92.72728 | 97.57576 | 96.36364 |
| | | std | 5.495137 | 7.624934 | 7.56559 | 6.474141 | 3.007572 | 4.305759 | 2.994475 | 3.366692 | 1.237811 | 3.271359 | 1.951147 | 1.156172 | 7.925269 | 3.149186 | 2.777316 | 2.099453 |
| | C3T | trial 1 | 59.09091 | 88.63636 | 96.59091 | 96.59091 | 33.21596 | 59.38967 | 70.77465 | 77.93427 | 49.08537 | 78.04878 | 83.84146 | 88.41463 | 81.25 | 97.32143 | 99.10714 | 99.10714 |
| | | trial 2 | 60.22727 | 84.09091 | 92.04546 | 94.31818 | 32.277 | 58.4507 | 70.53991 | 77.58216 | 51.21951 | 78.96342 | 84.60366 | 88.87195 | 90.17857 | 96.42857 | 99.10714 | 99.10714 |
| | | trial 3 | 68.18182 | 86.36364 | 90.90909 | 95.45454 | 31.69014 | 55.98592 | 66.90141 | 72.65258 | 53.35366 | 79.2683 | 85.36585 | 88.71951 | 81.25 | 96.42857 | 98.21429 | 100 |
| | | avg | 62.5 | 86.36364 | 93.18182 | 95.45454 | 32.39437 | 57.9421 | 69.40532 | 76.05634 | 51.21951 | 78.76017 | 84.60366 | 88.71951 | 84.22619 | 96.72619 | 98.80952 | 99.40476 |
| | | std | 4.953296 | 2.272725 | 3.006535 | 1.136365 | 0.769651 | 1.75795 | 2.171627 | 2.952993 | 2.134145 | 0.634658 | 0.762195 | 0.264034 | 5.154912 | 0.515493 | 0.515487 | 0.515493 |

Figure 6: **Full results:** across all trials for Table 1. Values that we used are highlighted and best performance of each method within supervised and UMA are bolded.

Table 6: **UMA performance compared to supervised baselines Using Train Method 1:** Each method is modular and can be decomposed to perform in the traditional supervised setting, or can be combined into ST, CA or CT3 to perform UMA. We developed all models from scratch, however, ST and CA resemble existing methods whereas C3T introduces novel mechanisms. Top-1 and Top-3 accuracies are reported for each dataset. Although ST performs poorly, it performs significantly better than randomly guessing, indicating it is still learning action information from the RGB data without any labels.

| | | UTD-MHAD | | CZU-MHAD | | MMACT | | MMEA-CL | |
|---|---|---|---|---|---|---|---|---|---|
| | Model | Top-1 | Top-3 | Top-1 | Top-3 | Top-1 | Top-3 | Top-1 | Top-3 |
| Supervised | IMU | **87.9** | **97.7** | **95.1** | 98.2 | 70.0 | 90.0 | 65.8 | 87.6 |
| | RGB | 53.8 | 73.1 | 94.0 | **99.7** | 42.1 | 61.6 | 54.2 | 77.1 |
| | Fusion | 62.5 | 82.2 | 95.0 | 98.5 | **76.7** | **92.0** | **80.1** | **92.7** |
| UMA | Random | 3.7 | 11.1 | 4.6 | 16.6 | 2.9 | 8.6 | 3.1 | 9.4 |
| | ST | 12.9 | 24.6 | 41.1 | 61.9 | 17.6 | 34.7 | 9.9 | 22.7 |
| | CA | 38.6 | 56.1 | 81.0 | 95.5 | 27.3 | 45.6 | 42.3 | 62.1 |
| | C3T | **62.5** | **86.4** | **84.2** | **96.7** | **32.4** | **57.9** | **51.2** | **78.8** |

and pass it into the RGB module. We concatenate the IMU data to provide a 60-channel input as the IMU modality and use depth as the input modality. Given the controlled environment and dense IMU streams, the models performed the best on this dataset.

**MMACT** The MMAct dataset (Kong et al., 2019) is a large scale dataset containing about 1,900 sequences of 35 action classes from 40 subjects on 7 modalities. This data is challenging because it provides data from 5 different scenes, including sitting a desk, or performing an action that is partially occluded by an object. Furthermore, the data was collected with the user facing random angles at random times. The dataset contains 4 different cameras at 4 corners of the room, and it measures acceleration on the user's watch and acceleration, gyroscope and orientation data from a user's phone in their pocket. We only use the cross-view camera 1 data, and again we concatenate the 4 3-axis inertial sensors into one 12 channel IMU modality.

**MMEA-CL** The Multi-Modal Egocentric Activity recognition dataset for Continual Learning (MMEA-CL) (Xu et al., 2023) is a recent dataset motivated by learning strong visual-IMU based representations that can be used for continual learning. It provides about 6,4000 samples of synchronized first-person video clips and 6-channel accelerometer and gyroscope data from a wrist worn IMU for 32 classes. The dataset's labels feature realistic daily actions in the wild, as opposed to recorded sequences in a lab. Due to issues with the data and technical constraints, we downsize the data proportionally from each class and use about the first 1,000 samples. Nonetheless, CT3's superior performance shows how this method can generalize to a different camera view, and different types of activities.

A.5   MAIN TABLE WITH ALL TRAIN METHOD 1:

Before the ablations discovered that Train method 4 (combined loss) was better for CA we used method 1 (Align first). This table shows the original experiment with all UMA models trained using method 1 (align first) and we observe no difference in the resulting rankings of the method.

## A.6 Additional Visualizations

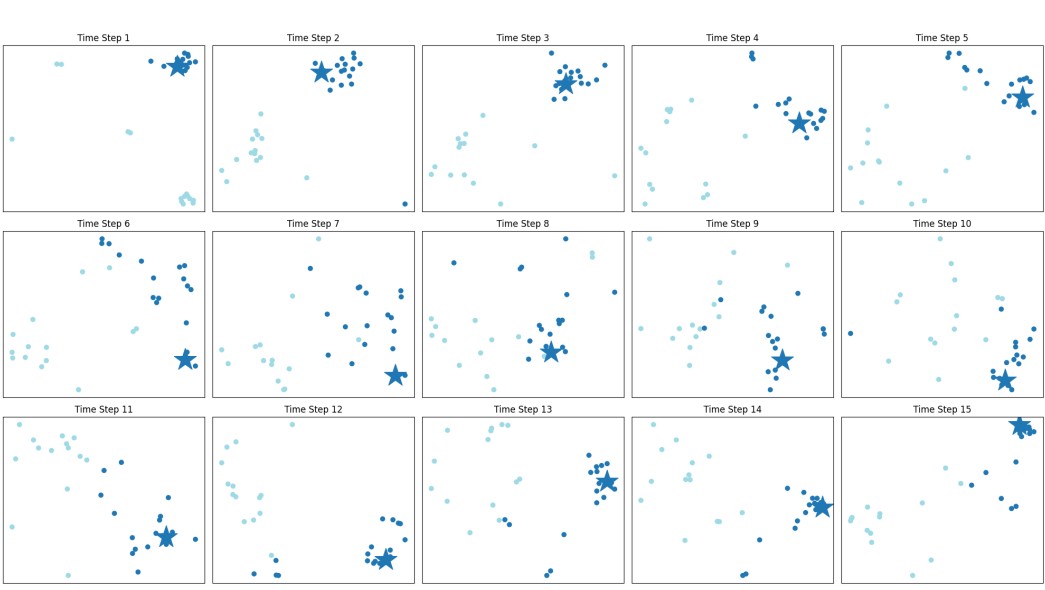

Figure 7: **C3T TSNE plot with shifted input:** This plot visualizes the TSNE plots of the $t_{\text{rec}} = 15$ latent $z_i = 0 \ldots t_{\text{rec}}$ for various points of two classes. The $z$s shown are the fused representation between the IMU and RGB modalities. The dark blue star is a point that was *shifted by 50%* compared to the rest of the visualized points for the class. Notice how some time steps are more distinctive than others between the classes, and the star tends towards the edge of the group for many $z$s.

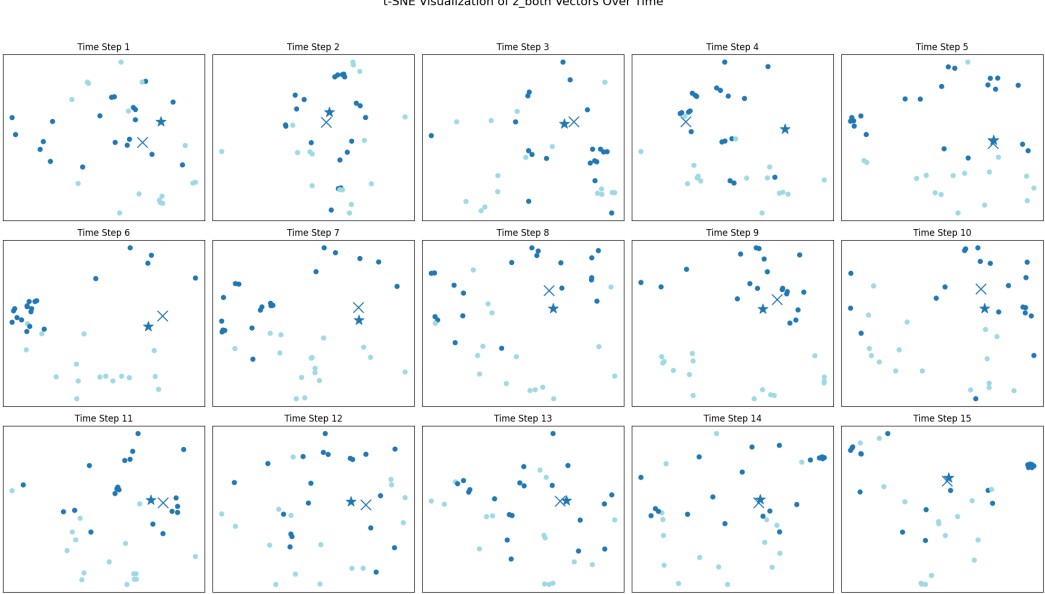

Figure 8: **C3T TSNE plot with shifted input:** Another TSNE plot with shifted input for different classes from the previous figure (Dark blue shows 'Swipe left' and light blue shows 'Pick up and throw'). This time X marks the regular input and star marks the shifted inputs. Unexpectedly, they are relatively close through out, indicating the temporal convolutions that constructed these z's might be doing part of the work when accounting for robustness to shift noise. Furthermore, notice how $z_7$ to $z_{10}$ indicate latent variables that are more distinctive. This could indicate that these time steps are most important for distinguishing between the two given classes.

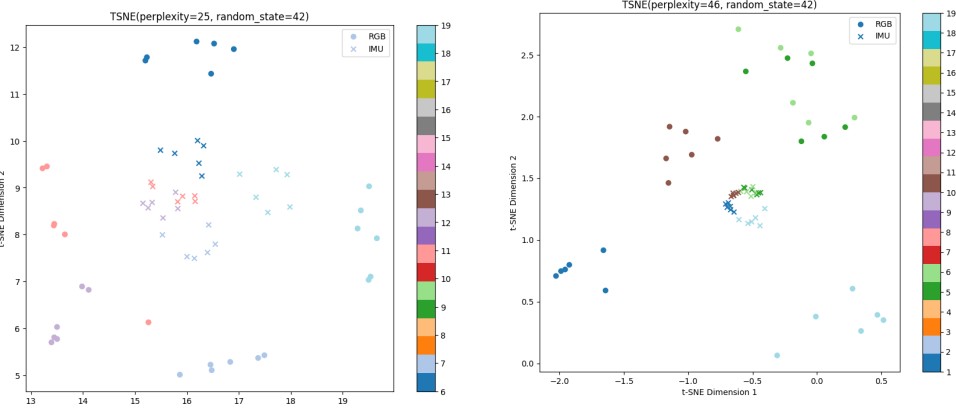

Figure 9: **CA TSNE on CZU Dataset:** These plots indicate the same trend discussed in the main paper, that the IMU points in the multimodal representation space tend to cluster in the middle and mirror the RGB points on the outside. Particularly, for the CZU dataset the IMU signals are stronger (60 IMU channels, 6 on 10 wearable devices) and this clustering tends to be stronger.

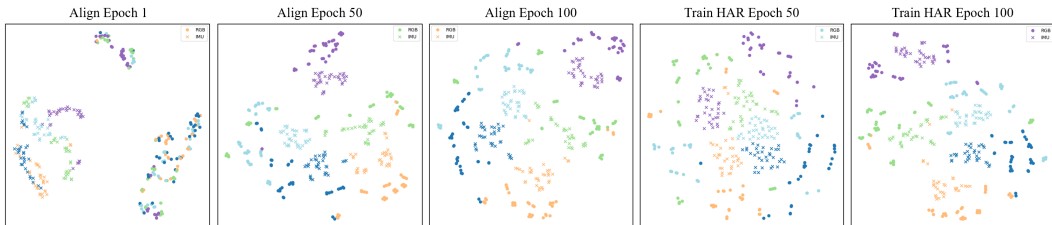

Figure 10: **CA TSNE Plots:** These are similar to the plot shown in the main paper. The following shows the progression of the latent representations for 5 classes (5 different colors) during training CA on the UTD-MHAD dataset in UMA. Circles indicate RGB data, and crosses indicate IMU data points.

Table 7: **UMA Testing on Each Modality** Accuracy MMACT (Kong et al., 2019) when trained for UMA and tested on only RGB data, only IMU data, or Both.

| Model | 1. RGB | 2. IMU | 3. Both |
|-------|--------|--------|---------|
| ST    | 25.8%  | 16.7%  | **25.8%** |
| CA    | 39.9%  | 26.9%  | **41.3%** |
| C3T   | 39.2%  | 31.7%  | **47.3%** |

# B   APPENDIX: ADDITIONAL EXPERIMENTS

## B.1   BASELINES

This method attempts to adapt existing methods to UMA and compare them as baselines against our methods.

Many works deal with robustness to missing sensor data during training or testing, however, few works deal with zero-labeled training data from one modality. As a result, constructing baselines is tricky and most methods had to be modified or adapted to fit our our approach. Even so, as shown in Table 8 These methods perform very poorly in the UMA setting.

We would like to note that all the baselines and methods were trained and tested on the data splits, i.e. it's not the case that they have different train a) and train b) splits. We believe this allows for fair comparison. In addition, the supervised baselines were only trained on the Train a) Supervised HAR split. This assures that the supervised baselines don't have an unfair advantage of seeing more data (they also only see 40% of the label data not 80%). The exact data splits are also provided in a separate repository linked in our code.

### B.1.1   SENSOR FUSION BASELINES

Sensor fusion is often broken down into the following 3 methods based on where the data are combined (Ramachandram & Taylor, 2017; Majumder & Kehtarnavaz, 2020; Sharma et al., 1998), also shown in Figure 11:

1. Early or data-level fusion combines the raw sensor outputs before any processing.

2. Middle/intermediate or feature-level fusion combines each sensor modality after some preprocessing or feature extraction.

3. Late or decision-level fusion combines the raw output, essentially ensembling separate models.

Many IMU-RGB based sensor fusion models have the ability to train on partially available or corrupted data and are robust to missing modalities during inference (Islam et al., 2022; Islam & Iqbal, 2020). No works have attempted the extreme case where one modality is completely unlabeled during training. Existing sensor fusion methods can be adapted to our setup using a psuedo- labeling technique, similar to the student-teacher model above. The difference here is that the model learns a

Table 8: **UMA with Existing Methods:** Most methods fail to adapt to zero-shot cross-modal transfer from the RGB to IMU sensor modalities. Imagebind performs well on MMEA, which is an eogecentric dataset, similar to Ego-4d in which Imagebind was trained on.

| Model | UTD-MHAD | MMACT | MMEA- CL | CZU-MHAD |
|---|---|---|---|---|
| Sensor Fusion (2019) (Wei et al., 2019) | 5.2% | 3.2 % | 4.1 % | 4.5 % |
| HAMLET (2020) (Islam & Iqbal, 2020) | 4.6 % | 3.2 % | 4.1 % | 4.5 % |
| ImageBind (2023) (Girdhar et al., 2023) | 11.3 % | 4.6 % | 40.1 % | 4.54 % |
| Student Teacher (Ours) | 12.9 % | 17.6 % | 9.9 % | 41.1% |
| Contrastive Alignement (Ours) | 38.6 % | 27.3 % | 42.3 % | 81.0 % |
| Contrastive Alignment Through Time (Ours) | **62.5** % | **32.4** % | **51.2** % | **84.2** % |

joint distribution between the two modalities so hopefully it may be able to learn some correlation between the models. Nonetheless, we show that these methods cannot generalize to the scenario where there is zero-labeled training data for one modality.

Let $g(\cdot, \cdot) : (\mathcal{X}^{(1)}, \mathcal{X}^{(2)}) \to \mathcal{Y}$. Our approach uses $\mathcal{D}_{HAR}$, to train by passing in zeros for one modality, e.g. we train $g(\cdot, \mathbf{0}) : \mathcal{X}^{(1)} \to \mathcal{Y}$. Then, with $\mathcal{D}_{Align}$ we use $g(\cdot, \mathbf{0})$ to generated psuedo-labels and then train $g(\mathbf{0}, \cdot, )$ with those labels.

We reproduced the conventional sensor fusion models (early, feature, and late) from (Wei et al., 2019) and indicate the performance of the top model on 8. We further reproduce a self-attention based sensor fusion appraoch (HAMLET (Islam & Iqbal, 2020)) and tested it on our setup. We follow a very similar architecture; however, extract spatio-temporal results using 3D convolution in the video as opposed to an LSTM and show similar results on the standard sensor fusion problem. We selected these model due to their state-of-the-art performance on the UTD-MHAD dataset, making them ideal benchmarks for comparison with our model.

To verify the integrity of our reproduced models we compared to state-of-the-art reported methods and showed similar performance results. The results are given in Table 9 These experiments prove that our sensor fusion baselines are comparable to SOTA method, and they fail to perform well in the UMA setting underscoring the importance and novelty of our work.

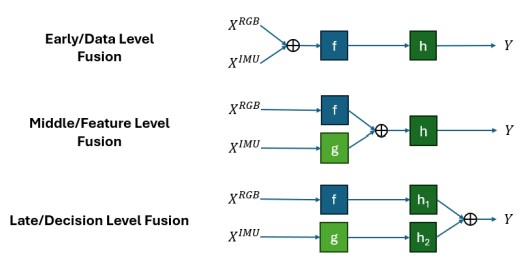

Figure 11: Types of Sensor Fusion

Table 9: SOTA Sensor Fusion Performance on UTD-MHAD. † (Wei et al., 2019)

, ∗ (Islam & Iqbal, 2020)

| REPORTED MODELS | ACCURACY |
|---|---|
| HAMLET ∗ | 95.12% |
| WEI ET AL. † | 95.6% |
| REPRODUCED FROM † | ACCURACY |
| EARLY FUSION | 86.71% |
| FEATURE FUSION | 95.60% |
| LATE FUSION | 94.22% |

### B.1.2 CONTRASTIVE LEARNING BASELINE

ImageBind (Girdhar et al., 2023) learns encoders for 6 modalities, (Images/Videos, Text, Audio, Depth, Thermal and IMU) by performing CLIP's training method (Radford et al., 2021) between each of those encoders and the Image/Video encoder. It was well tested for image, text and audio based alignment, retrieval and latent space generation tasks, however was not well test with IMU data and not used for specific tasks, such as HAR. In addition, one fundamental difference between Imagebind and CA is that Imagebind constructs a latent space amongst the sensing modalities and text and aligns between them. We hypothesize that this is vector space is more difficult and unnecessary to construct, for human action recognitoin using sensing modalities. The text modality, although sequential in nature, does not have a time dimension, thus it cannot leverage correlations between modalities in time like C3T.

Let's denote the video, IMU and text encoders as $g^{(1)} : \mathcal{X}^{(1)} \to \mathcal{Z}, g^{(2)} : \mathcal{X}^{(2)} \to \mathcal{Z}$, and $g^{(3)} : \mathcal{X}^{(3)} \to \mathcal{Z}$ respectively. We perform two conventional task-specific adaptations for CLIP models. First, we attempt zero-shot transfer, in which we pass all the action labels through the text encoder. For a dataset with $C$ classes, we have $\hat{Z}^{(3)} = (\hat{z}_1^{(3)} \ldots \hat{z}_C^{(3)})$. Finally, for a given IMU sample $(x_i^{(2)}, y_i) \in \mathcal{D}_{Test}$, we pass $x_i^{(2)}$ through the IMU

encoder $g^{(2)}$ and retrieve $\hat{z}^{(2)}$. Finally, we classify the point by looking at which points gives the highest cosine similarity score in the latent space, e.g. $\hat{y}_i = argmax_j \frac{\langle x_i^{(2)}, \hat{z}_j^{(3)} \rangle}{||\langle x_i^{(2)}, \hat{z}_j^{(3)} \rangle||}$.

Given that ImageBind is a large model trained on massive corpuses of data it becomes impractical to train the model from scratch on our smaller datasets collected from wearables and edge devices. Instead, we fine-tuned the ImageBind model using a linear projection head on the encoders, that can then be trained for a specific task. The results of this method are depicted in Table 8.

The results show a poor generalization of Imagebind to most experiments on our setup, and we hypothesize a few reasons. Firstly, ImageBind is a large model and may either overfit to small datasets, or not have enough training examples to learn strong enough representations. Second, ImageBind was pre-trained on Ego4D and Aria which contain egocentric videos to align noisy captions with the IMU data, whereas our datasets had fixed labels and were mostly 3rd person perspective. In fact ImageBind performed the best on the one egocentric dataset we used, MMEA-CL(Xu et al., 2023). Lastly, Imagebind was trained on a IMU sequences of 10s length sampled at a much higher frequency, thus we zero-padded or upsampled the IMU data to fit into ImageBind's IMU encoder, and the sparse or repetitive signal may have been too weak for ImageBind's encoder to accurately interpret the data.

## B.2 FEW SHOT CROSS-MODAL ADAPTATION

***How quickly can our models learn from labeled IMU samples?***

As illustrated in Figure 12, CA demonstrates faster learning, reaching peak performance within 20 shots, while C3T requires about 40 shots. Neither method matches the supervised IMU performance of 87.9% reported in Table 1, but they approach the fusion performance of approximately 62%.

It's important to note that this comparison with the supervised baselines may not be entirely fair, as the supervised baselines had access to the entire Train HAR dataset (40% of the data), whereas, the few-shot learning was conducted on the validation set (10% of the data). Given that the supervised IMU and fusion models share the same architecture as CA, repurposed for the supervised setting, we would expect similar performance under equal conditions.

Nonetheless, these experiments clearly demonstrate CA's superior ability in few-shot cross-modal learning compared to C3T. As shown in Figure 12, both IMU and combined modality performance improve with increasing shots, while RGB performance remains constant due to the learning shots containing only labeled IMU data.

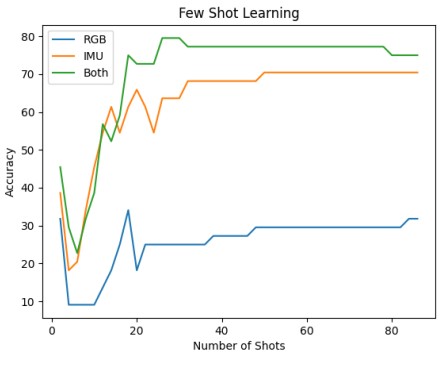 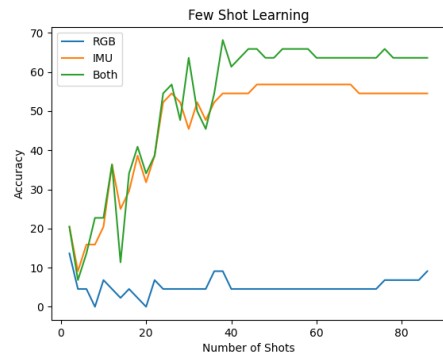

(i) **CA Cross-Modal Few Shot**    (ii) **C3T Cross-Modal Few Shot**

Figure 12: **Cross-Modal Few Shot Learning Comparison:** (a) CA method and (b) C3T method performance in cross-modal few shot learning scenarios when testing on IMU, RGB and Both modalities. RGB performance remains the same because the learning shots only contain labeled IMU data. However, we can see IMU and Both performance rise.

## B.3 ADDITIONAL ABLATIONS:

We conducted a brief ablation study on the alignment loss, comparing our cosine similarity approach with the conventional L2 loss. As shown in Table 10, the results strongly support our initial intuition presented in Section 3. The substantial performance gap between cosine similarity and L2 loss for both CA and C3T

models underscores that cosine similarity is indeed a more effective measure of alignment for high-dimensional vectors in this context. These findings align with well-established principles in high-dimensional space analysis, reinforcing the validity of our approach. Given the well-established nature of these results, we have included this comparison in the supplementary material rather than the main paper, focusing the primary discussion on novel contributions.

Table 10: **Alignment Loss Comparison:** Performance of CA and C3T models using Cosine Similarity and L2 loss for alignment on the UTD-MHAD dataset.

| Model | Cosine Similarity | L2 Loss |
|-------|-------------------|---------|
| CA    | **44.32**         | 2.27    |
| C3T   | **62.50**         | 3.41    |

Table 11: **Architecture Ablation Extensions:** This shows an extension of our architecture ablations to baselines. It shows a comparison of different architectures for RGB and IMU encoders across various methods. We report encoder types for spatial and temporal dimensions of RGB data, and the temporal dimension for IMU data, along with the number of parameters (in millions) and accuracy for each configuration. Convolutional architectures generally yielded superior performance, while still maintaining a relatively low model size. These results indicate that C3T's performance advantage stems from its methodological approach rather than solely from it's attention head or size.

| Method | RGB | | IMU | Params (M) | Accuracy (%) |
|--------|-----|--|-----|-----------|--------------|
| | Spatial | Temporal | Temporal | | |
| ST | Conv | Conv | Conv | 129.2 | **12.9** |
| | Conv | Conv | Attn | 97.8 | 10.2 |
| | Conv | Attn | Conv | 871.2 | 11.4 |
| | Attn | Conv | Conv | 291.5 | 5.7 |
| CA | Conv | Conv | Conv | 163.8 | **38.6** |
| | Conv | Conv | Attn | 132.3 | 19.3 |
| | Conv | Attn | Conv | 905.7 | 34.1 |
| | Attn | Conv | Conv | 326.0 | 26.1 |
| C3T | Conv | Conv | Conv | 137.7 | **62.5** |
| | Conv | Conv | Attn | 106.3 | 15.9 |
| | Conv | Attn | Conv | 879.6 | 53.4 |
| | Attn | Conv | Conv | 300.0 | 33.0 |
| FUSION | Conv | Conv | Conv | 163.8 | 62.5 |
| | Conv | Conv | Attn | 132.3 | 77.3 |
| | Conv | Attn | Conv | 905.7 | **89.8** |
| | Attn | Conv | Conv | 326.0 | 64.8 |
| IMU | – | – | Conv | 32.0 | **87.9** |
| | – | – | Attn | 0.5 | 27.3 |
| RGB | Conv | Conv | – | 97.3 | 53.8 |
| | Conv | Attn | – | 839.2 | **71.6** |
| | Attn | Conv | – | 259.5 | 64.8 |

