# OpenReview forum: "C3T: Cross-modal Transfer Through Time for Human Action Recognition"
_ICLR.cc/2025/Conference — ICLR 2025 Conference Withdrawn Submission_

### Official Review · Reviewer_tUtt · 2024-10-29

**Soundness:** 1
**Presentation:** 1
**Contribution:** 2
**Rating:** 5
**Confidence:** 3

**Summary:**

The authors explore a method for transferring knowledge between modalities by utilizing a unified multimodal representation space for Human Action Recognition (HAR). Through Unsupervised Modality Adaptation (UMA), they designed Student-Teacher (ST), Contrastive Alignment (CA), and Cross-modal Transfer Through Time (C3T).

**Strengths:**

In this multimodal era, utilizing multimodal action features for recognizing human actions is an interesting research topic. The main challenge of this multimodal approach is transferring discriminative action features from one modality to another. Additionally, the alignment between two modalities presents another challenge in an unsupervised context. The authors are trying to address this challenge in human action recognition.

**Weaknesses:**

- The writing and organization of the paper is poor, it needs to be reorganized to clearly present the main contributions of the paper. Additionally, some of the sentences are not clear and need to be revised. For example, line 018 and 269 “C3T removes the MLP layer at the end of the feature …”.

-  Figure 1 does not clearly show the motivation for the proposed approach, specifically how unsupervised domain adaptation is performed in the figure and how discriminative features from one modality transfer to other modalities as high-level representations.

- Some of the mathematical representations are not correct, for example, line 299.

- The main weakness of the paper is its contribution. The proposed approach is essentially a comparison with the Student-Teacher (ST), Contrastive Alignment (CA), and Cross-modal Transfer Through Time (C3T) methods. As far as I understand, the main contribution of the paper is the design of cross-modal transfer learning for an unlabeled modality, which is not clearly presented. Additionally, it is unclear how the proposed approach transfers representative action features from one domain to another through a simple contrastive learning design.

- Lines 267-307 present the main contributions of the paper and achieve good results, as mentioned by the authors. However, this section needs a clearer presentation of the proposed approach, including how they formulated the method and how they transfer discriminative action features from one modality to another. Additionally, the authors mention alignment between modalities, it would be helpful to specify what kind of alignment techniques they utilized or proposed.

**Questions:**

see the weaknesses

---

### Official Review · Reviewer_TdGJ · 2024-10-30

**Soundness:** 1
**Presentation:** 2
**Contribution:** 2
**Rating:** 3
**Confidence:** 4

**Summary:**

The authors explained that they developed three methods to perform UMA: Student-Teacher (ST), Contrastive Alignment (CA), and Cross-modal Transfer Through Time (C3T). They performed extensive experiments on four datasets to illustrate the contribution of the presented methods.

**Strengths:**

The paper exhibits the experimental results of three methods and performs an ablation study for specific settings using the presented method.

**Weaknesses:**

1. The paper writing has lots of problems.
2. Explanation for presented methods is not clear.
3. Experiment comparison is not reliable since they do not compare with related works.

**Questions:**

After carefully read the paper, some suggestions are listed here,
1）Unsupervised Modality Adaptation is frequently used for cross-modal learning in many applications. It cannot be regarded as a new contribution of this paper. Referring to “Development and comparative analysis of three methods”, Student-Teacher (ST), the definition is incomplete and inaccurate. It is suggested to consider refining the key innovations and redefining the key representations for innovation.

2) In Table 2, there have Train a) and Train b). It is not easy to contact to the
It is suggested to add explanation for them in caption or main text.

3) In figure 3, the model size refers Bubble size or hidden size? It is suggested to use one definition in the caption.

4) In experiment section, the proposed approach is not compared with related frameworks. Why? There are many existing works as listed in Section 2.

5) Compared to ST and CA, the Cross-modal Transfer Through Time (C3T) performs pseudo label generation and alignment in temporal sequences. Other operations, including feature learning, RGB-IMU alignment, classification, also follow the same procedures. So the novelty of proposed method need to reconsider.

6) “We believe ST performs the worst since the student is bound by the performance of the teacher.” It is suggested to give a judgment following experiment results, but not subjective assumption.

7) Writing problems:
We formalize and explore an understudied cross-modal transfer setting we term Unsupervised Modality Adaptation (UMA),
 zero labeled instances of the test modality are available during training
We develop three methods to perform UMA: Student-Teacher (ST),

---

### Official Review · Reviewer_C89t · 2024-11-02

**Soundness:** 2
**Presentation:** 2
**Contribution:** 2
**Rating:** 5
**Confidence:** 4

**Summary:**

The authors investigate methods of transfer learning between modalities using unsupervised modality adaption. Three methods were developed, namely  Student-Teacher (ST), Contrastive Alignment (CA), and Cross-modal Transfer Through Time (C3T). The results showed that C3T outperformed the other techniques by 8% and nears the supervised setting

**Strengths:**

The authors conducted extensive experiments on four datasets. The authors verified the proposed model qualitatively and quantitatively.

**Weaknesses:**

1-The paper needs polishing. I found some typos that need modifications. For example:
* Line 13-15 "We formalize and explore an understudied cross-modal transfer setting we term Unsupervised Modality
Adaptation (UMA), where the modality used in testing is not used in supervised
training, i.e. zero labeled instances of the test modality are available during training" --> I believe this sentence need to be rewritten.
*Line 209 --> It should be f^2 not f^1
*Line 210 -->LCE(Pf2(x), Pf2(x)) one of them should be LCE(Pf2(x), Pf1(x))
*Line 400 --> TSNE--> tSNE
2-The contribution of C3T is not clear
3- The architecture of the attention module in C3T is not clear at all.
4- tSNE needs reference.

**Questions:**

1- The authors utilized ResNet18 for video representation. Could you consider 3D convolution Models including I3D?
2- It is not clear what t represents does not the whole sequence or a group of frames.
3- What is the size of t_{rec} because it manages the number of losses.
4-  Could you make IMU supervised learned and RGB is the model that is unlabeled? Especially the IMU has higher performance.
5-  The author stated in lines 410-417 "Interestingly, we observed that IMU data points consistently cluster towards the center of the plot,
with RGB points surrounding them. This pattern persists even in early alignment stages, suggesting it’s not solely due to labeled RGB HAR training. While this might indicate that RGB data is more informative, it contradicts our quantitative findings where supervised
IMU models outperform RGB models for our given datasets. " However the figure show that IMU data samples are are compact than RGB data samples. Actually, It obvious that IMU should have higher performance. Could you elaborate? I need your comment on my observation.
6- Does the tSNE visualize training or testing data? if it is training could you show us testing data.

---

### Official Review · Reviewer_peSv · 2024-11-04

**Soundness:** 2
**Presentation:** 2
**Contribution:** 2
**Rating:** 3
**Confidence:** 4

**Summary:**

The paper investigates an unsupervised modality adaptation method for the action recognition model. The objective is to transfer the knowledge from a modality with a labeled dataset, to another modality with an unlabeled dataset. Three methods are studied, namely Student-Teacher, Contastive Alignment (CA), and the proposed Cross-modal Transfer Through Time (C3T). The key in C3T is to do the alignment on each temporal step, in contrast to CA which aligns on video-level feature representation. The experiments are conducted on 4 RGB-IMU datasets, and shown consistent improvement on C3T compared to the other methods.

**Strengths:**

1. The paper investigates a practical problem, which is unsupervised cross-modal knowledge transfer, that is increasingly relevant due to easier access to multi-modal sensors.
2. The proposed C3T gives consistent improvement to ST and CA methods.

**Weaknesses:**

1. No clear comparison with state-of-the-arts in the main body of the paper, even though 4 different public datasets are used.
2. The explanation for the C3T method can be improved by translating the sentences into simpler mathematical expressions.
3. The novelty is minor, which is only on doing the alignment per temporal step, without proposing a new alignment method, or an architectural change to current SoTAs. The lack of SoTA comparison also makes it hard to justify the significance of the novelty besides its improvement upon ST and CA.

**Questions:**

1. The clarity of the paper can be improved, such as table and text placement, and typos (e.g. line 438, name of Chapter 4).
2. Table 1 needs more clarity for better readability, addressing points such as:
- Can you clarify more on the "random guessing" method that is used here?
- The modalities used in the UMA experiments

---

### Note · Authors · 2024-11-15

I have read and agree with the venue's withdrawal policy on behalf of myself and my co-authors.